# Respiratory Telerehabilitation of Boys and Young Men with Duchenne Muscular Dystrophy in the COVID-19 Pandemic [note 1]

**DOI:** 10.3390/ijerph18126179

**Published:** 2021-06-08

**Authors:** Agnieszka Sobierajska-Rek, Łukasz Mański, Joanna Jabłońska-Brudło, Karolina Śledzińska, Eliza Wasilewska, Dominika Szalewska

**Affiliations:** 1Department of Rehabilitation Medicine, Faculty of Health Sciences with Institute of Maritime and Tropical Medicine, Medical University of Gdansk, 80-219 Gdansk, Poland; jjbrudlo@gumed.edu.pl (J.J.-B.); dominika.szalewska@gumed.edu.pl (D.S.); 2Department of Physical Therapy, Faculty of Health Sciences with Institute of Maritime and Tropical Medicine, Medical University of Gdansk, 80-211 Gdansk, Poland; lmanski@gumed.edu.pl; 3Department of Internal and Pediatric Nursing, Faculty of Health Sciences with Institute of Maritime and Tropical Medicine, Medical University of Gdansk, 80-211 Gdansk, Poland; karolina.sledzinska@gumed.edu.pl; 4Department of Pulmonology and Allergology, Faculty of Medicine, Medical University of Gdansk, 80-211 Gdansk, Poland; ewasilewska@gumed.edu.pl

**Keywords:** DMD, children, telerehabilitation, respiratory rehabilitation

## Abstract

Background: The COVID-19 pandemic forced reorganization of the multidisciplinary healthcare system for Duchenne muscular dystrophy. Digital solutions seem to be optimal for providing rehabilitation at this time. The aim of this study was to investigate whether it is possible to conduct respiratory physical therapy with the use of telerehabilitation in Duchenne muscular dystrophy. Methods: The study was conducted during an online conference for families with DMD. During the physical therapy panel we showed the video with the instructions of respiratory exercises. All participants (*n* = 152) were asked to fill in the online survey evaluating the quality, acceptance, and understanding of the instructions. Results: The survey was filled in by 45 (29.6%) participants. The mean rating of satisfaction was 4.70/5, and for intelligibility was 4.78/5. Thirty-seven (82.2%) patients declared that they had performed the exercises, all caregivers declared that it was possible to perform the proposed exercises a few times a week or daily, and only two respondents replied to invitations to individual online sessions. Conclusions: Findings from the study show that respiratory telerehabilitation may be implemented for DMD patients; however, the interest in digital rehabilitation among caregivers of DMD boys in Poland is low. The reasons for this situation require further research.

## 1. Introduction

Duchenne muscular dystrophy (DMD) is the most common and severe neuromuscular disease, affecting approximately 1 in 3500 persons worldwide. This incurable X-linked recessive disorder is caused by a mutation in the dystrophin gene, leading to impaired production or function of the dystrophin protein, which, in turn, leads to progressive muscle weakness. Boys with DMD in their second decade of life lose ambulation, and then the ability to stand or change position. The milestones in the disease progression are: lack of ambulation; loss of ability to use upper limbs; gastrointestinal problems, resulting in malnutrition; heart failure; and respiratory insufficiency [1,2].

Historically, the cardiorespiratory system failure caused death before 18 years of age. Today, there is still no cure for DMD, but the appropriate treatment and medical multidisciplinary care can improve the quality of life of affected boys, and extend their life expectancy to more than 30 years of age [2,3].

Assessment and treatment of respiratory system dysfunction is one of the most important problems for DMD patients. Deteriorating muscle strength and tension may lead to progressive scoliosis. Massive deformity raises the risk of cardiopulmonary dysfunction [2,4]. The first sign of respiratory failure is obstructive sleep apnea (OSA), which causes chronic headaches, daytime drowsiness, episodes of distraction, and difficulty in falling asleep and waking up [5].

The respiratory muscle weakness leads to secondary changes, such as decreased lung compliance, ineffective cough with deterioration of airway clearance, and repeated infections [3,4]. Many studies have shown that respiratory function declines at a rate of 6–11% annually in patients with DMD [1,6,7]. A major component of respiratory dysfunction seems to be a decline in inspiratory muscle strength [6,8].

Therefore, rehabilitation with a particular emphasis on respiratory training is recommended as one of the key elements of DMD patients’ care [1,3]. Until now, the rehabilitation procedures have been offered in specific, dedicated rehabilitation institutions open for patient visits, or else rehabilitation interventions were performed by physical therapists at home—usually for patients with an advanced stage of the disease.

It is well documented that rehabilitation with glucocorticosteroid treatment slows down the disease progress; it prolongs ambulation, delays respiratory function deterioration, and decreases the risk of severe scoliosis. Traditional rehabilitation consisting of stretching, positioning, exercises improving strength and endurance, postural re-education, and respiratory management is a key intervention for maintaining and preserving optimal function and participation [9]. Constant direct treatment implemented by physical therapists and occupational therapists should be tailored to the patient’s age, individual needs, and stage of disease [10].

The COVID-19 pandemic has influenced health care systems across the entire world. Due to the fact that DMD patients are at risk of severe complications from COVID-19 [11], the multidisciplinary healthcare team is seeking the safest way to provide constant care and treatment at the highest possible level. The innovative digital form of physical therapy, which is easily available, free of charge, and adjustable, can support the daily routine of home training during the time of social distancing [9].

The purpose of this study was to investigate whether or not it is possible to conduct respiratory physical therapy via telerehabilitation in boys with Duchenne muscular dystrophy. The study also aimed to assess the acceptance of asynchronous telerehabilitation methods in this group of patients.

## 2. Materials and Methods

### 2.1. Study Design

In this multicenter study, the possibility of implementing respiratory exercises via online video program instruction into home-based rehabilitation routines in Duchenne muscular dystrophy patients was evaluated. The study was conducted as a part of the Multidisciplinary Care Program for Patients with Duchenne Muscular Dystrophy at the Rare Disease Centre (RDC), University Clinical Centre, Medical University of Gdańsk, Poland. The University Clinical Centre is a member of the TREAT NMD Alliance Neuromuscular Network. This study was conducted from December 2020 to February 2021, and was the first part of the project “E-monitoring of pulmonary function in patients with Duchenne muscular dystrophy undergoing respiratory rehabilitation at home”.

Approval for the study was obtained from The Committee of Ethics no. NKBBN/260/2021, which conformed to the principles embodied in the Declaration of Helsinki.

### 2.2. Participants

The study population included boys with DMD diagnosis based, according to the guidelines, on the presence of typical clinical symptoms, genetic testing, and/or muscle biopsy results [1]. All patients with families are members of the Parent Project Foundation in Poland [12]. The patients were recruited during the online conference “DMD–Let’s be together” for Polish families with DMD organized by the Rare Disease Centre and the Parent Project in January 2021. Details of the medical status of the patients were collected from a survey sent online to each participant of the conference (see Section 2.4).

### 2.3. Telerehabilitation Exercises

The video with the instructions for respiratory exercises—positive inspiratory pressure, glossopharyngeal breathing, and positive expiratory pressure—was shown to parents and patients at the conference during the physical therapy panel. Participants were asked to perform the exercises at home without physiotherapists’ assistance. Patients were encouraged to train every day, three times a day. On the video, patients received following instructions:Breath stacking (positive inspiratory pressure): Patients were asked to take a maximal inhale, and then, without exhaling, to take another 1–3 breaths, and hold for 5 s. Boys were instructed to repeat the exercise 5 times. The aim of the exercise was to increase lung volume. The same exercise was presented with the use of balloons; patients were instructed to inhale the air from a balloon, without exhaling, and try to hold as much air in the lungs as possible for 5 s, repeating 5 times.Glossopharyngeal breathing: Patients were instructed to push a series of small volumes of air with their tongue and pharynx into the lungs by saying ”cat” in Polish. Patients were encouraged to repeat the exercise 5 times.Positive expiratory pressure: Inflating the balloon. Patients were instructed to inflate the balloon by inhaling by nose and exhaling by mouth, repeating the exercise 3 times. Caregivers were encouraged to try different types of balloons in order to find optimally elastic material (optimal resistance).

The video is still available on the Internet at https://www.youtube.com/watch?v=AEaxOsuJimU&feature=youtu.be (accessed on 28 April 2021).

### 2.4. Data by Questionnaire

After the conference, all participants (*n* = 152) were asked to fill in the online survey (Google Forms) evaluating this home-based program. The questionnaires were completed in Polish through an online survey platform. One month after the conference, participants who did not respond to the questionnaire were asked to take part in the survey again.

The structured questionnaire consisted of questions that covered several areas: (1) demographic data; (2) respiratory exercises; (3) additional information about exercises and the home-based program; and (4) program satisfaction assessment.

Demographic data consisted of questions about age and ambulation according to the 10-point Vignos scale (VS) (the scores on the VS range from 1 to 10; “1” means that the subject can walk and climb stairs without assistance, while “10” means that the subject is confined to a bed). The VS allows the monitoring of the course of the disease, and focuses on functional activities, mainly of the lower limbs, in which deterioration is considered as important milestone in the progression of the disease. The VS is commonly used to determine functional dependence in neuromuscular diseases [13]. Upper limb functional status was assessed with the 6-point Brooke scale (BS) (“1” means that the patient can abduct their arms in a full circle until they touch above their head, while “6” means that the patient has no useful function of the hands) [14].

Respiratory exercises: The participants were asked: (a) whether the exercises were performed (yes/no; alone/with assistance); (b) about the possibility of implementing this treatment into their daily routine, and about how often the exercises could be performed from the patient’s perspective (3 times daily/1–2 daily/a few times a week/not possible); and (c) about the difficulty of the exercises (all exercises possible to perform/possible to perform after practicing/some exercises too difficult/all exercises too difficult).

Additional information: The caregivers were invited to contact us via e-mail, in case they needed more detailed individual consultation or training.

Satisfaction from the program: Respondents were also asked to express their general satisfaction and views on the appropriateness and intelligibility of the exercises on a 6-point scale, where “0” meant the worst and “5” meant the best score.

### 2.5. Statistical Analysis

Analyses were conducted using Statistica 13.0 (Statsoft, Krakow, Poland). Descriptive statistics are presented as means and standard deviations (SDs), or medians and interquartile ranges (IQRs). Spearman’s correlation was used to investigate the correlation between non-normal distributed data of age and the Vignos scale, between age and the Brooke scale, and between general satisfaction and the abovementioned functional scales. To determine differences between the respondents regarding the difficulty and frequency of the exercises, the Kruskal–Wallis test was used. *p*-Values of less than 0.05 were considered significant.

## 3. Results

### 3.1. Participants Characteristics

Finally, out of 152 participants, only 45 (29.6%) filled in the online survey. The mean age of the patients was 11.00 (SD 7.88) years old; 27 (60.0%) patients were ambulant. A significant correlation was observed between patients’ age and ambulatory status (R = 0.56, *p* = 0.000), and between age and upper limb functional status (R = 0.64, *p* = 0.000). Detailed ambulatory and upper limb functional status is presented in Table 1. The individual age, VS Score, and BS scale of each participant are presented in the Appendix A.

### 3.2. Exercises

The video with respiratory exercises was displayed 164 times during the first month.

#### 3.2.1. Performance of the Exercises

The number of participants who tried to perform the exercises by themselves was 8 (17.7%), 29 attempted the exercises with assistance (64.4%), while 8 (17.7%) did not perform the exercises (Figure 1).

#### 3.2.2. Possibility and Frequency

Apart from one, all respondents (*n* = 44) declared that it was possible to implement the respiratory exercises to their home-based rehabilitation routine in a full or modified version.

Most of the respondents (30 patients) declared the possibility of performing the exercises every day. Details of the frequency of the exercises are presented in Figure 2.

Detailed characteristics of the declared frequency of the exercises are presented in Table 2.

#### 3.2.3. Difficulty

Only 5 (11.1%) respondents declared that they were able to perform all of the exercises; 22 (48.9%) respondents declared that they were able to perform the exercises correctly, but that they needed to practice; 16 (35.6%) respondents declared that they were able to perform the exercises after some modifications; and 2 (4.4%) patients declared that it was too difficult to perform the exercises. There was no significant difference among the patients’ abilities to perform the exercises regarding age, VS, or BS functional status. Detailed characteristics related to ability of performing the exercises are presented in Table 3.

### 3.3. Additional Consultations

Only three caregivers responded to the invitation for further individual consultation or assistance with exercise performance.

One respondent—a non-ambulatory 18-year-old patient—chose phone consultation. The patient and his parents asked questions about symptoms of dyspnoea, eventual disturbing symptoms regarding respiratory condition demanding hospitalization, and the efficacy of home use of a pulse oximeter. Parents were also worried because of the cancellation of regular multidisciplinary consultations caused by the COVID-19 pandemic. However, they admitted that they did not try to perform the proposed respiratory exercises.

Another respondent chose e-mail consultation—the mother of a 6-year-old ambulant boy reported that her son had difficulties with the respiratory exercises and performing pulmonary function tests because of his inability to perform forced expiration.

### 3.4. Satisfaction, Appropriateness, and Intelligibility

The ratings varied from 5 to 3. The mean general satisfaction rating of the online respiratory program was 4.70/5 (SD 0.63), the mean appropriateness rating was 4.60/5 (SD 0.72), and the mean intelligibility rating was 4.78/5 (SD 0.47).

Characteristics of patients who gave a rating of 3 for general satisfaction, appropriateness, and intelligibility are presented in Table 4.

There were no correlations observed between general satisfaction rating and age (*p* = 0.66), VS (*p* = 0.536), nor BS (*p* = 0.641). There was also no relationship revealed between the appropriateness rating and the abovementioned variables (*p* = 0.58; *p* = 0.83; *p* = 0.11, respectively). Moreover, a lack of correlation was also observed between intelligibility rating and age, VS, or BS (*p* = 0.11; *p* = 0.90; *p* = 0.89 respectively).

## 4. Discussion

The presented study provides insight into the possibility of implementing respiratory exercises as part of home-based rehabilitation routines for 2.5–24-year-old, male, Caucasian DMD patients at different stages of the disease. We used asynchronous telerehabilitation methods, i.e., the video with the instruction of respiratory exercises addressed to DMD patients. This method has been recently recommended due to the SARS-CoV-2 virus pandemic.

The most important finding is the fact that the telerehabilitation is acceptable by patients with DMD and caregivers, and can be performed using video techniques as instructions. This fact is especially important during the pandemic period, as our previous study showed, in which almost all patients stopped institutional physical therapy due to the national recommendations of social isolation, and the majority of them continued physical therapy at home [9]. Limited attendance for institutional rehabilitation indicates the urgent need to implement well-designed telerehabilitation programs that meet the specific needs of DMD patients.

Interestingly, before the COVID-19 pandemic period, in 2015, the American Thoracic Society (ATS) and the European Respiratory Society (ERS) suggested performing telerehabilitation in order to increase the availability of pulmonary rehabilitation to wider groups of patients [15]. These guidelines have now become of exceptional importance due to the potential to reduce the risk of infection and transmission of the SARS-CoV-2 virus.

Current studies concerning respiratory telerehabilitation report that this intervention is effective in reducing disability and improving quality of life, and may reduce the risk of severe exacerbations. This form of intervention is widely used in the COPD population [16,17,18]. Moreover, there is evidence that telerehabilitation in the pediatric population may be as effective as face-to-face treatment [19]. Tanner et al., in their study conducted during the time of the COVID-19 pandemic, reported that pediatric rehabilitation can feasibly be provided through telerehabilitation technologies, and families report high levels of satisfaction with this model of care. However, they noticed limiting factors for implementation, such as limited pay or reimbursement, perceived and actual technological barriers, liability concerns, and privacy concerns [20].

The second finding of this study is that most patients performed the exercises with their caregiver’s assistance. This observation may provide a hint for the future design of online interventions, that the presence of a caregiver during the session might be necessary. Promising findings show that the majority of respondents declared that it would be possible to exercise more frequently than 1–2 times a day; however, the question of the effectiveness of the home-based respiratory exercises in this group of patients remains open. The observations of other researchers confirm that the effectiveness of home-based respiratory exercises is highly related to the frequency of performance. They indicate that such interventions should be implemented at least twice or three times daily [21,22].

The study group was heterogeneous regarding the functional condition of the upper limbs: there were patients without any functional limitations, and there were some with no useful function (BS 1–6). The findings revealed that even patients with the most severe condition were able to perform the exercises; however, they reported the need of some modifications. The research did not reveal any significant differences concerning age, VS, nor BS among patients with different levels of ability to perform the exercises. From this observation, it can be stated that the level of acceptability of the exercises is independent of the stage of the disease; thus, the exercises can be implemented for both ambulatory and non-ambulatory patients. In the study group there were no patients confined to their beds; thus, it is impossible to assess whether this program is feasible for this group of patients.

In the presented study, it was observed that the youngest patients (mean age 4.0) declared that the exercises were too difficult to perform or needed some modifications; due to the small sample size, this finding needs to be interpreted with caution; however, the establishment of specific respiratory exercises for DMD boys of preschool age should be considered. It is worth emphasizing that teaching proper breathing patterns to activate all respiratory muscles, as well as to preserve good chest mobility, should be an element of recreational activities from the earliest years after diagnosis. Creating good habits of breathing is much easier when there is no visible weakness of respiratory muscles and no respiratory function deterioration. Moreover, preschool children, as well as boys with autism spectrum disorders or concentration disorders, may have difficulties with understanding and performing the exercises. In these children, combination of in-person treatment in the rehabilitation center with the home-based program may be more beneficial.

The program of respiratory rehabilitation used in this study was designed on the basis of guidelines and recommendations of respiratory exercises for neuromuscular diseases to increase lung volume and activate respiratory muscles [23,24,25]. Results of the study indicate that, for 40% of respondents, some or all of the exercises were too difficult to perform. Due to this observation, the aim of our future study is to modify the exercises to be less difficult for that group of patients. Moreover, the problem of objective assessment of respiratory function that can be effectively performed by the DMD patient at home still remains unsolved, and is under investigation.

Another finding is the low response rate to the online survey conducted after the conference and to the invitation for individual consultation, as well as the low number of patients who made an effort to perform the proposed exercises.

Our previous study concerning digital physical therapy showed a similarly low response rate to the online physical therapy workshops. Moreover, the survey also showed that more than half of the DMD patients included in the abovementioned study felt overloaded with home schooling responsibilities [7]. This may be the reason for the low response to the survey and/or treatment provided online. Another reason may be related to insufficient trust in healthcare interventions conducted indirectly. It should be mentioned that telerehabilitation is a new method in Poland, and is not widely known among parents/caregivers of DMD patients.

Similarly, low response rates are described in telerehabilitation studies involving patients with different disorders. The study of Hensen concerning pulmonary telerehabilitation of 134 patients with COPD revealed that only 12.19% of the group that met the inclusion criteria managed to complete the intervention protocol [16].

A limitation of the study is the relatively small group of subjects. Moreover, the assessment was only based on respondents’ declarations; thus, we were not able to avoid bias in our assessment of upper limb function and ambulatory status.

However, these are still interesting findings that may change everyday practice, drawing clinicians’ attention to the possibility of using online video methods and conducting exercises by the patients themselves at home. To the best of our knowledge, this is the first study concerning digital respiratory rehabilitation in DMD boys.

In summary, today, the multidisciplinary DMD healthcare team is facing the challenge of ensuring consistency in care and treatment. Respiratory rehabilitation is one of the most pressing problems. It must be emphasized that in-person contact with physical therapists experienced in respiratory rehabilitation and assessment of neuromuscular diseases should always be an essential element of standard care of DMD patients [26].

Therefore, online rehabilitation during the COVID-19 pandemic, as well as conventional care of DMD boys, should always rely on an individual approach, taking into consideration patients’ age, functional abilities, cognitive limitations, and levels of cooperation. This study shows ratings of the quality and understanding of the instructions, as well as acceptance of the proposed exercise program, indicating that it is possible to implement such intervention in daily home-based rehabilitation routines.

## 5. Conclusions

The study shows that respiratory telerehabilitation may be implemented in DMD patients; however, it should include caregivers’ assistance. The exercises should be adapted to the specific needs and the age of the patients. The interest in digital rehabilitation among caregivers of DMD boys in Poland is low, which requires further research. Moreover, there is a need for well-designed randomized clinical trials assessing the effectiveness of the proposed interventions.

## Figures and Tables

**Figure 1 ijerph-18-06179-f001:**
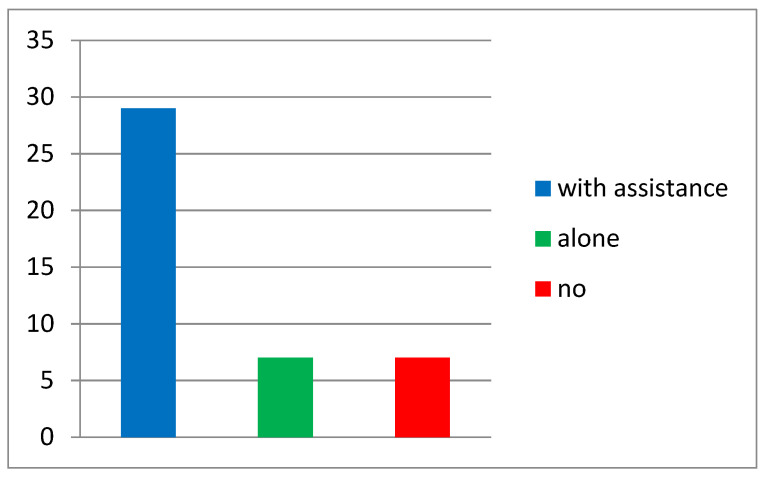
Number of participants who performed the exercises.

**Figure 2 ijerph-18-06179-f002:**
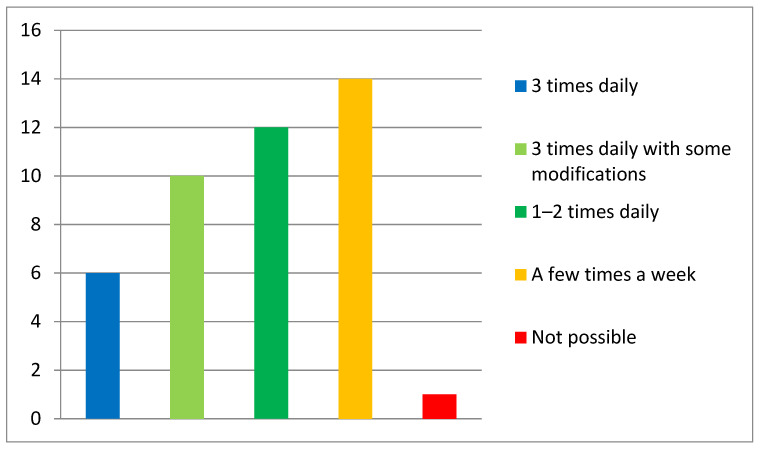
Declared frequency of the exercises.

**Table 1 ijerph-18-06179-t001:** Detailed ambulatory (VS) and upper limb functional status (BS).

Ambulatory Status VS	N = 45 (%)	Upper Limb Functional Status BS	N = 45 (%)
Walks and climbs stairs without assistance.	9 (20)	Can abduct the arms in a full circle until they touch above the head.	25 (55.5)
Walks and climbs stairs with railing or assistance.	12 (26.7)	Can raise arms above head only by flexing the elbow.	12 (26.7)
Walks and climbs stairs slowly with aid or railing (more than 25 s for 8 steps).	4 (8.9)	Cannot raise hands above head, but can raise a glass of water to the mouth.	3 (6.7)
Walks unassisted, rises from chair, cannot climb stairs.	2 (4.4)	Cannot raise hands to the mouth but can use hands to hold a pen or pick up coins from the table.	3 (6.7)
Uses a wheelchair.	18 (40)	Cannot raise hands to the mouth and has no useful function of hands.	2 (4.4)

**Table 2 ijerph-18-06179-t002:** Description of patient groups with different frequency of exercises.

	Able to Perform 3 Times Daily	Able to Perform 1–2 Times Daily	Able to Perform a Few Times a Week	Not Able to Perform	*p* Value *
N = 45	18	12	14	1	
Mean age (SD)	12.83 (5.9)	11.83 (5.52)	8.25 (4.89)	4	0.08
Median VS (IQR)	3 (8.0)	6 (7.0)	2 (2.0)	2	0.51
Median BS (IQR)	1 (2.0)	2 (1.0)	1 (1.0)	2	0.39

VS: Vignos scale; BS: Brooke scale. Details of these scales are described in Section 2.4. * Kruskal–Wallis test.

**Table 3 ijerph-18-06179-t003:** Description of patient groups with different levels of exercise performance results.

	Able to Perform All Exercises	Able to Perform after Practicing	Able to Perform after Modifications	Not Able to Perform	*p* Value *
N = 45	5	22	16	2	
Mean age (SD)	12.83 (5.9)	11.83 (5.52)	8.25 (4.89)	4	0.50
Median VS (IQR)	3 (1)	2.5 (7)	2.5 (7)	1.5 (1)	0.59
Median BS (IQR)	1 (0)	1 (1)	2 (1)	1 (0)	0.30

VS: Vignos scale; BS: Brooke scale. Details of these scales are described in Section 2.4. * Kruskal–Wallis test.

**Table 4 ijerph-18-06179-t004:** The lowest ratings of satisfaction, appropriateness, and intelligibility.

	Satisfaction	Appropriateness	Intelligibility
Number of participants	4	6	1
Mean age (SD)	7.25 (4.97)	7.66 (4.40)	21.0
Median VS (IQR)	2.5 (4.0)	3.0 (1.0)	9.0
Median BS (IQR)	1.0 (0.0)	1.0 (0.0)	1.0

## Data Availability

The data presented in this study are available on request from the corresponding author. Full data are not publicly available due to privacy restrictions.

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
