# Peer review of "Respiratory Telerehabilitation of Boys and Young Men with Duchenne Muscular Dystrophy in the COVID-19 Pandemic"

_ijerph, 2021, doi:10.3390/ijerph18126179_

Round 1

Reviewer 1 Report

Agnieszka Sobierajska-Rek et al., have performed a study on the feasibility of the respiratory telerehabilitation  in the DMD patients. This study is important for the DMD patients community. The manuscript is well written and the conclusion drawn from the result is appropriate considering the study design. However, there are many caveats in the study and the presentation of the result needs improvement.

My comments are below

  1. The authors should describe the result in the section 3.4 (Satisfaction, appropriateness and intelligibility ) in the form of table.
  2. The author should provide the  individual age, VS score and BS scale of each 45 participant as a supplementary table. 
  3. The author should mention whether any of these patients have been doing the respiratory exercise in person in the rehabilitation center before they are being told for telerehabilitation exercise.
  4. Were they doing only the respiratory  or other exercises during the telerehabilitation?
  5. Did the author find any difference in the overall effect of respiratory exercise done during telerehabilitation and in person in the rehabilitation center?
  6. Did the author  analyze the respiratory parameters of these patients before the start and end of the telerehabilitation?
  7. In the discussion section, the author should also describe about the importance of respiratory exercise in person in the rehabilitation center. 

Reviewer 2 Report

The purpose of the study was to investigate whether it is possible to conduct respiratory physical therapy with the use of telerehabilitation in boys with Duchenne 7muscular dystrophy. The aim of the study was also to assess the acceptance of asynchronous telerehabilitation method in this group of patients.

    The article is well written. I would like to recommend its publications. There are some minor comments:

First, the authors are suggested to provide the benefit of tranditional rehabilitation for patients with DMD in the introduction.

Second, have the authors conducted sample size calculation before the study? If so, please provide.

Third, the percentage of the participants with specific ambulation status should be provided.

Reviewer 3 Report

Dear authors

Although rarely, well-written articles with few defects occur.
This article is one of them. The weaknesses are the small sample of partecipants recruits (described in any case within the limitations section) and the lack of calculation of the sample size.
The statistics are well conducted, and the materials and methods are well described.
Being a cross sectional survey, the conclusions cannot be directly used to impact clinical practice but can open up food for thought on future research areas of DMD.
It could help to insert a table with the characteristics of the partecipants recruits (e.g. age, sex, etc.) and analysis of the results by subgroup (for example gender or age groups) and write the name of the survey platform used.

All these data, however, do not change the discussions, the rusilati and the intrinsic value of the paper, which remains well written but with a low impact on the clinical practice. 

Best Regards

Reviewer 4 Report

In line 19, in the summary, you should not speak in the first person, but in the impersonal. The same happens on line 220, 222, 259, 261, 267, 284, 287, 299, 304, 311. In line 52, the text is missing the bibliographic reference.
In line 56, these bibliographic references can be put like this (3-5).
The number of bibliographic references is deficient.    

Round 2

Reviewer 1 Report

Tha authors have addressed all my concerns and now the quality of manuscript has improved significantly.